# Synergistic dual anion regulation unlocks giant thermopower and power density in hydrogel

Hongbing Li [1], Zhangjie Gu[1], Yaling Zhu[1], Zhaoyang Jiao[1], Jinya Tian[1], Yi Li[1], Yongping Chai[1] & Xiaodong Chi [1,2] ✉

Harvesting low-grade heat from the environment and converting it into electricity holds the potential to power devices independent of cables or batteries. However, their effectiveness is limited by weak ion selectivity and insufficient concentration gradients. Here, we introduce the use of a calix[4]pyrrole as effective anion traps to selectively capture $Fe(CN)_6^{4-}$ and $Cl^-$ anions, enabling simultaneous modulation of redox ion distribution and suppression of anion mobility under a temperature gradient. This strategy combines desolvation-induced entropy gain with thermodiffusion enhancement arising from the mobility asymmetry between cations and anions. This leads to a synergistic boost in thermopower to an impressive $8.1\,mV\,K^{-1}$, and results in a 20-fold increase in output power compared to the $PVA/Fe(CN)_6^{3-/4-}$ system. Demonstrated through a proof-of-concept wearable device with 36 unipolar elements, our system generated nearly 3 volts under ambient conditions. This strategy offers a promising route toward thermoelectric materials with enhanced thermopower for efficient harvesting low-grade thermal energy.

Generating electricity from low-grade heat has been of interest for many decades. Widespread low-grade heat sources ($< 100\,°C$) are typically derived from the environment, industrial processes, the human body heat, and solar heating[1,2]. However, the efficient transformation of this ubiquitous but underutilized energy form faces challenges due to the limited efficacy of existing technologies[3–5]. Thermoelectric (TE) technology, particularly valued for its ability to convert thermal energy directly into electrical power, encounters limitations with conventional semiconductor-based devices, which typically demonstrate low thermopower (in microvolts per kelvin, $\mu V\,K^{-1}$), alongside issues related to fragility, cost, and complex manufacturing that restrict their application in flexible and wearable technologies[6–11]. Thermodiffusion cells (TDCs) and thermogalvanic cells (TGCs) present a promising pathway for converting low-grade heat directly into electricity, showcasing enhanced thermopower in millivolts per kelvin ($mV\,K^{-1}$) and providing a scalable solution for energy conversion[12–14]. TDCs operate on the principle of the Soret effect, where a temperature gradient induces the thermodiffusion of cations and anions, creating a polarized voltage difference. Although TDCs exhibit considerable thermopower, the intermittent nature of their electrical output constrains their practical application. In contrast, TGCs facilitate the generation of consistent electric power across varying temperature gradients via electrochemical redox reactions at two electrodes[15,16]. Nonetheless, the relatively lower thermopower of TGCs compared to that of TDCs, coupled with the potential for electrolyte leakage, poses challenges that necessitate further advancements[17,18].

Thermopower in thermoelectric materials is closely related to the entropy difference ($\Delta S$) between redox ions and their concentration gradient across a thermal gradient, with efforts predominantly focused on enhancing $\Delta S$ to augment thermopower due to the concentration difference ($\Delta C$) typically being negligible[19,20]. Strategies to increase $\Delta S$ involve altering the solvation shells of the redox couple or the redox couple's structure itself. For instance, Zhou et al. enhanced

[1]State Key Laboratory of New Textile Materials and Advanced Processing, School of Materials Science and Engineering, Huazhong University of Science and Technology, Wuhan, China. [2]Shenzhen Huazhong University of Science and Technology Research Institute, Shenzhen, China. ✉e-mail: xchi@hust.edu.cn

the thermopower of an aqueous $Fe(CN)_6^{3-}/Fe(CN)_6^{4-}$ electrolyte from 1.4 to 4.2 mV K$^{-1}$ by adding guanidinium and urea, leveraging ionic bonding to reorganize the solvation shells, thereby increasing $\Delta S$[1]. Similarly, Kim et al. found that mixing methanol with water as a solvent rearranged the solvation shells, boosting the thermopower to 2.9 mV K$^{-1}$ [13]. However, such methods can diminish the electrolyte's ionic conductivity, potentially impacting power performance.

Increasing $\Delta C$ represents an alternative strategy to enhance thermopower. Li et al. generated a continuous $\Delta C$ of $Fe(CN)_6^{3-}$ and $Fe(CN)_6^{4-}$ ions through an in-situ induced photocatalytic process[20]. Yamada et al.'s created a substantial $\Delta C$ of $I^-/I_3^-$ at the cold electrode with the selective encapsulation of $I_3^-$ in α-cyclodextrin[21]. Zhou et al. further amplified both $\Delta C$ and $\Delta S$ by inducing the crystallization of $Fe(CN)_6^{4-}$ ions with guanidinium cations, achieving a thermopower of 3.73 mV K$^{-1}$ [19]. Despite these advancements, thermopowers resulting from thermogalvanic effects of redox couples remain modest compared to those derived from ionic thermodiffusion, highlighting the critical need for integrated strategies that can synergistically couple and amplify both effects to achieve substantial gains in thermoelectric performance[22–28].

Herein, we introduced a approach using calix[4]pyrrole to develop a quasi-solid-state thermocell that not only overcomes the electrolyte leakage issue but also significantly enhances thermopower to 8.1 mV K$^{-1}$ and power density to 5.33 mW m$^{-2}$ K$^{-2}$. The integration of phenylboronic acid-functionalized calix[4]pyrrole (C4P) within the hydrogel matrix substantially enhances both thermogalvanic and thermodiffusion effects through precise anion modulation. Acting as an anion-selective molecular trap, C4P captures $Fe(CN)_6^{4-}$ and $Cl^-$ through stabilizing hydrogen bonds within its macrocyclic structure[29]. This selective host-guest complexation induces desolvation of $Fe(CN)_6^{4-}$, thereby increasing redox entropy ($\Delta S$) and establishing a pronounced concentration gradient ($\Delta C$) across the temperature gradient. Selective suppression of anion mobility, coupled with

preserved high cation mobility, amplifies the asymmetry of ionic mobility. Together, these synergistic effects maximize the contributions of both thermogalvanic and thermodiffusion processes, delivering performance in ionic thermoelectric systems.

## Results

### Cell preparation and thermoelectric properties

Gel-based thermoelectric cells (TECs) were prepared using polyvinyl alcohol (PVA), employing sequential freeze-thaw cycles and ion exchange techniques based on the traditional gel thermoelectric cell design (PVA/$Fe(CN)_6^{3-/4-}$)[12]. When subjected to a modest temperature gradient of 10 K, these TECs exhibited two distinct voltage response profiles (Fig. 1b). Unlike in liquid-based systems, ion thermodiffusion within the gel matrix cannot be neglected, as it significantly impacts the voltage stabilization time due to charge accumulation asymmetry at the hot and cold interfaces. The pure PVA/$Fe(CN)_6^{3-/4-}$ gel exhibited a thermopower comparable to its aqueous counterpart (~1.4 mV K$^{-1}$), indicating a negligible contribution from thermodiffusion ($S_{td}$), and reached voltage equilibrium within about 10 min the, the shortest time among all tested samples. In comparison, systems relying solely on the thermomigration of $K^+$ and $Cl^-$ ions required longer to stabilize. The introduction of C4P alone significantly boosted thermopower but prolonged the equilibrium time to nearly 20 min, attributed to the reduced mobility of $Fe(CN)_6^{4-}$ ions due to host–guest complexation with C4P. Systematic testing of six TECs prototypes revealed that the combined incorporation of KCl and C4P into the PVA/$Fe(CN)_6^{3-/4-}$ matrix notably enhanced thermopower from 1.4 mV K$^{-1}$ to 8.1 mV K$^{-1}$ (Fig. 1c, top). This enhancement exceeded the additive effects of KCl or C4P alone and is attributed to two key factors: (1) the selective complexation of $Fe(CN)_6^{4-}$ ions by C4P generates a redox ion concentration gradient and suppresses the redox activity of the $Fe(CN)_6^{4-}$ anion, promoting the $Fe(CN)_6^{3-} + e^- \rightarrow Fe(CN)_6^{4-}$ reaction and thus increasing thermopower; and (2) C4P restricts the mobility of anions ($Fe(CN)_6^{4-}$

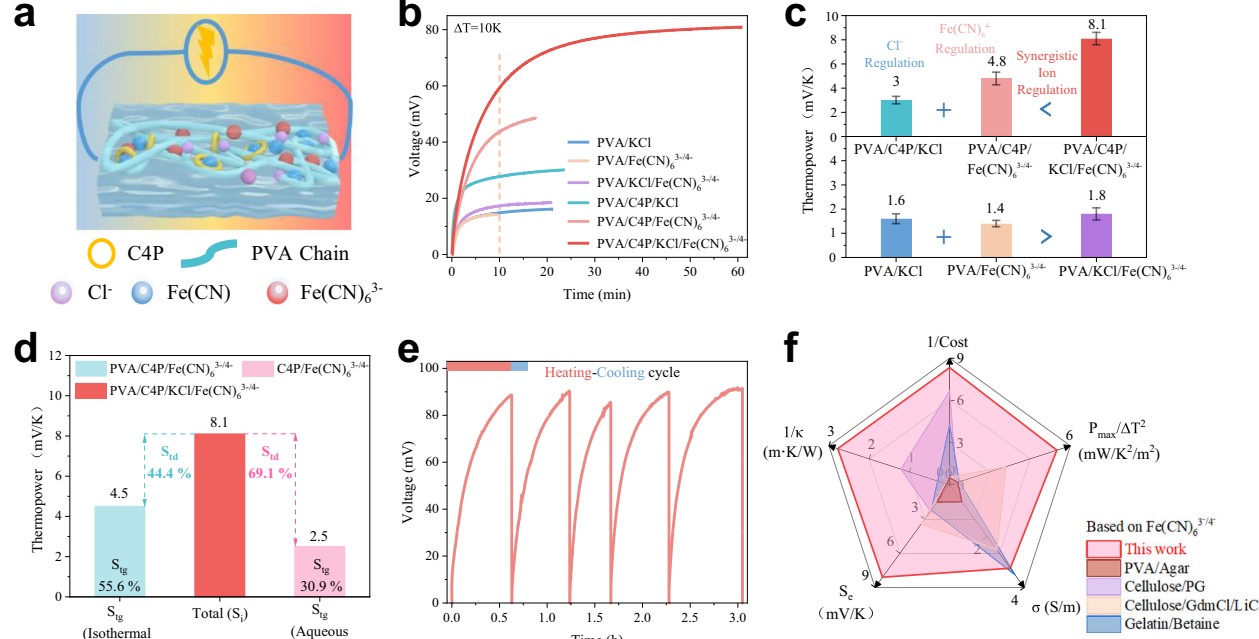

**Fig. 1 | Thermoelectric properties of gel TECs. a** Schematic diagram of the synergistic dual anion regulation-enhanced TEC. **b** Voltage-time profiles for gel TECs, measuring with gels of dimensions 1 × 1 × 1 cm, positioned 1 cm apart between the hot and cold ends under a controlled temperature differential (ΔT) of 10 K. **c** Comparison of the thermopower across various as-fabricated TECs. Data are presented as mean values ± SD (n = 3). **d** Decoupling of thermogalvanic

contribution to thermopower, determined by (left) isothermal electrochemical measurements using Pt/SCE electrodes and (right) direct thermopower measurement in aqueous solution with PVA replaced by deionized water. **e** Open-circuit voltage-time curves of PVA/C4P/KCl/$Fe(CN)_6^{3-/4-}$ TEC over five thermal cycles, ΔT = 11 K. **f** Five-dimensional performance comparison of the C4P-enhanced TECs with other TECs reported in the literatures (See supplementary Table 1).

and $Cl^-$ anions), thereby increasing the migration asymmetry between anions and cations and effectively amplifying the thermodiffusion effect. In contrast, KCl alone produced only modest thermopower enhancement, and the combined effect with $Fe(CN)_6^{3-/4-}$ falls short of their simple sum (Fig. 1c, bottom) because elevated ionic strength accelerates both cation and anion migration, diminishing the thermodiffusion potential. Thus, selectively impeding anion migration is essential for fully exploiting thermodiffusion in salt-doped gel systems.

A comparative study was conducted between PVA/KCl/$Fe(CN)_6^{3-/4-}$ and PVA/C4P/KCl/$Fe(CN)_6^{3-/4-}$ to further explore the effect of C4P's host-guest interactions with anions on the TECs performance (Supplementary Figs. 4 and 5). The results revealed that while C4P's inclusion slightly influence the thermal or ion conductivities of the cells (Supplementary Figs. 6 and 17), it did lead to a notably increase in thermopower. This enhancement resulted in higher short-circuit current density and a substantial increase in the maximum power density ($P_{max}$). Specifically, at a thermal gradient of 10 Kelvin, the $P_{max}$ for the TECs including C4P increased from $34.7\,mW\,m^{-2}$ to $532.6\,mW\,m^{-2}$ (Supplementary Fig. 5A). To decouple and quantify the individual contributions to the overall thermoelectric response, we first employed an isothermal binary-cell configuration (Supplementary Fig. 7). In the PVA/C4P/$Fe(CN)_6^{3-/4-}$ system, the thermopower of $Fe(CN)_6^{3-/4-}$ redox couple was determined to be $4.5\,mV\,K^{-1}$ (Supplementary Fig. 8), with thermogalvanic accounting for 55.6% of the total thermopower (Fig. 1d), closely matching the directly measured thermopower ($4.8\,mV\,K^{-1}$). As a complementary approach, we measured the thermopower of C4P/$Fe(CN)_6^{3-/4-}$ in a liquid-phase configuration, replacing the gel with deionized water under identical ionic composition and conditions (Supplementary Fig. 9). Here, thermodiffusion is negligible, and the measured value reflects a 30.9% thermogalvanic contribution. Together, the two methods consistently reveal the dual origins of the enhanced thermopower. Notably, thermodiffusion-derived values from both methods (3.6 and $5.6\,mV\,K^{-1}$, respectively) exceed the PVA/C4P/KCl control, due to increased $K^+$ mobility and C4P's selective suppression of $Cl^-$ and $Fe(CN)_6^{4-}$ transport, which amplifies mobility asymmetry and boosts overall thermopower. The operational stability was further tested using a homemade temperature control device over five thermal cycles, confirming the cells' robust thermal-electrical conversion efficiency (Fig. 1e).

A multidimensional performance-cost analysis was conducted to benchmark our approach against reported ionic thermoelectric strategies (Fig. 1f). In this comparison, the cost evaluation excluded common matrix and electrolyte components (PVA, $Fe(CN)_6^{3-/4-}$, and KCl) and considered only the materials responsible for enhancing thermoelectric performance. To ensure fair comparison, only literature reports employing carbon-based electrodes were included to eliminate electrode-dependent effects. The results demonstrate that our dual-anion regulation approach achieves a significantly higher thermoelectric enhancement than existing strategies, highlighting both the efficacy and practical potential of our molecular-level design. Additionally, we performed control tests using Au@Cu and Pt electrodes to assess the influence of electrode materials on thermoelectric performance (Supplementary Figs. 41–50). Although Au@Cu electrodes exhibited higher apparent thermopower than carbon paper (CP), subsequent analyses revealed that this enhancement primarily originated from corrosion-induced parasitic reactions. In contrast, both carbon and Pt electrodes delivered stable and intrinsic values, underscoring the importance of electrode selection for the reliable evaluation of ionic thermoelectric materials.

## Mechanism of the TEC

A working principle for the synergistic dual anion regulation enhanced thermopower of the system was proposed, as shown in Fig. 2a. Central to this design is the introduction of selective host–guest interactions

between C4P and $Fe(CN)_6^{4-}/Cl^-$, which act as a molecular gating mechanism for both ionic mobility and redox thermodynamics. Multiple lines of evidence confirmed that C4P preferentially binds $Fe(CN)_6^{4-}$ over $Fe(CN)_6^{3-}$, including NMR (Supplementary Figs. 12 and 13), Ultraviolet−visible (UV−vis) spectroscopy (Fig. 2b and Supplementary Figs. 18 and 19), X-ray diffraction (XRD; Supplementary Fig. 20), and X-ray photoelectron spectroscopy (XPS; Supplementary Figs. 21 and 22). When 1 equiv. of $Fe(CN)_6^{4-}$ was added to the C4P solution, a slight chemical shift change was observed. In contrast, no significant chemical shift change was noted for C4P/$Fe(CN)_6^{3-}$ (Supplementary Fig. 12). Additionally, UV−visible spectroscopy showed significant red shifts in the absorption peaks of $Fe(CN)_6^{4-}$ after C4P addition, while no obvious adsorption peaks shift of $Fe(CN)_6^{3-}$ (Fig. 2b). XPS analysis showed an obvious binding energy increasement of $Fe(CN)_6^{4-}$ after C4P addition, while no binding energy change observed in $Fe(CN)_6^{3-}$ (Supplementary Fig. 21). This selective anion recognition breaks the symmetry of the redox pair and enables asymmetric molecular-level control across the thermocells. By selectively immobilizing $Fe(CN)_6^{4-}$ and $Cl^-$ via host−guest complexation, C4P effectively reduces the thermal mobility of these anions under a temperature gradient while allowing $K^+$ to migrate freely (Supplementary Figs. 12 and 13). Cyclic voltammetry (CV) measurements confirmed a substantial decrease in overall ionic mobility after C4P addition (Supplementary Fig. 15). Molecular dynamics (MD) simulations further disentangle the individual contributions, revealing that C4P strongly suppresses $Fe(CN)_6^{4-}$ and $Cl^-$ transport, with $Cl^-$ mobility reduced by 91.3% (Fig. 2c and Supplementary Fig. 16). This pronounced asymmetry in ionic diffusivity amplifies the cation−anion mobility difference, thereby enhancing thermopower under the applied thermal gradient.

Beyond anions transport regulation, entropy modulation introduced by host−guest complexation plays a pivotal role in boosting the thermogalvanic component of the thermopower (Fig. 2d, e and Supplementary Figs. 24 and 25). At the cold end, C4P selectively binds $Fe(CN)_6^{4-}$, disrupting its hydration shell and releasing multiple structured water molecules into the bulk phase (Supplementary Figs. 31–34). This desolvation event dramatically increases the local entropy associated with $Fe(CN)_6^{4-}$, as confirmed by ITC thermograms and MD-derived entropy changes (Fig. 2f and Supplementary Fig. 23). The resulting entropy gain increases the value of $\Delta S$ in the thermopower relation $S_{redox} = \frac{\Delta S}{nF}$. The complexation step steepens the entropy difference between redox states, thus boosting thermopower. Meanwhile, CV measurements revealed diminished redox currents upon C4P binding, indicating suppressed electrochemical activity due to effective concentration reduction of the free $Fe(CN)_6^{4-}$ species (Supplementary Figs. 14 and 26). More importantly, this entropy modulation is temperature-responsive (Supplementary Fig. 28). At the hot end, elevated thermal energy weakens the C4P−$Fe(CN)_6^{4-}$ as evidenced by a declining association constant ($K_{as}$) with increasing temperature (Supplementary Fig. 27). This thermal weakening promotes complex dissociation, followed by rehydration of $Fe(CN)_6^{4-}$, effectively reversing the entropy gain achieved through desolvation at the cold end. This reversible entropy-switching mechanism enables a dynamic response to the applied thermal gradient: entropy is increased at the cold end through binding and desolvation, and decreased at the hot end through dissociation and rehydration (Supplementary Movies 1 and 2). Such a cyclical entropy change across the device mimics an entropy pump, which is highly favorable for enhancing the net thermogalvanic voltage.

Taken together, the entropy modulation effect, achieved through supramolecular desolvation and reversible binding, acts in concert with ion mobility regulation to deliver a dual-boosting strategy. This mechanism establishes C4P not just as a passive additive, but as an active thermodynamic regulator, capable of elevating ionic thermopower beyond conventional limits.

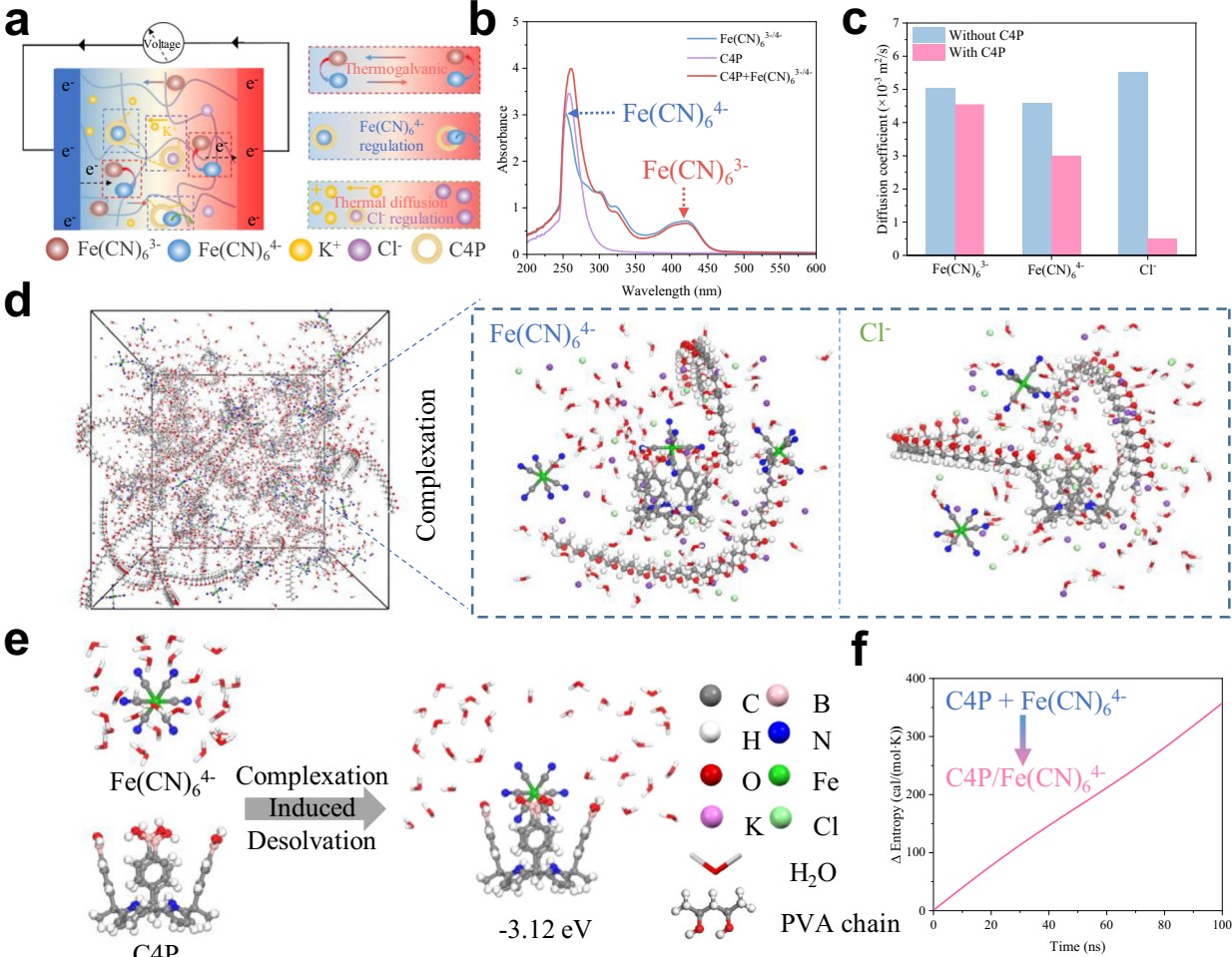

**Fig. 2 | Mechanism of the synergistic anions regulation. a** Schematic representation of ion diffusion, redox reactions, and host-guest interactions in PVA/C4P/KCl/Fe(CN)$_6^{3-/4-}$ under temperature gradients. **b** UV–vis spectra of individual components (C4P, Fe(CN)$_6^{3-/4-}$) and their mixtures. **c** Simulated changes in ionic mobility for Fe(CN)$_6^{3-}$, Fe(CN)$_6^{4-}$, and Cl$^-$ before and after C4P incorporation. **d** Spatial ion distribution in the gel from molecular dynamics simulations. The enlarged image shows the complexation of C4P with Fe(CN)$_6^{4-}$ and Cl$^-$. **e** Representative MD trajectory showing the stepwise complexation of Fe(CN)$_6^{4-}$ by C4P, accompanied by cavity encapsulation and partial desolvation. **f** Simulated entropy change associated with the stepwise complexation of Fe(CN)$_6^{4-}$ by C4P.

## Working modes of the gel TECs

The discharge behavior of the TEC with dual effects differs significantly from traditional thermogalvanic and TDCs (Fig. 3a). TGCs operate continuously, generating electrons through oxidation and reduction of ion pairs at the cold and hot electrodes, respectively. In contrast, TDCs function capacitive-like, featuring non-Faradaic discharge currents. To illustrate these differences, we examined three representative TECs (Fig. 3b). The power output profile of the PVA/Fe(CN)$_6^{3-/4-}$ based thermogalvanic cell stabilizes rapidly upon connection to an external circuit, maintaining relatively constant energy output despite a gradual and inevitable power reduction over prolonged discharge periods. On the contrary, the PVA/KCl-based thermodiffusion cell exhibited a rapid and continuous decline in power output, approaching zero after approximately 5 min (Fig. 3b). This rapid decrease occurred due to internal charge equilibration, requiring the system voltage to be periodically re-established under open-circuit conditions with a maintained temperature gradient. This behavior arises because the external circuit resistance (100 Ω) is substantially lower than the cell's internal resistance (~ 3 kΩ). When the external load resistance was increased to 5 kΩ, the discharge rate slowed, but the power still steadily diminished, approaching zero within approximately 1 h (Supplementary Fig. 29). These results clearly highlight that purely thermodiffusion-driven cells, although capable of generating high

initial peak power, are not sustainable for long-term continuous power delivery. As a result, the dual-effect TEC, incorporating both thermodiffusion and thermogalvanic effects, did not achieve immediate equilibrium upon connection to an external circuit (Fig. 3c). Instead, the voltage gradually decreased over extended discharge times. However, unlike purely diffusion-driven cells, this dual-effect TEC maintained a relatively stable and sustained power output due to the continuous conversion of ion pairs within the cell. As a result, it effectively avoided the rapid energy depletion typical of pure thermodiffusion systems. Comparative analysis of power densities clearly demonstrate that the dual-effect TEC achieves long-term stability comparable to a conventional thermogalvanic cell, with significant additional contributions from the thermodiffusion effect. Consequently, the integrated system yields a remarkable 20-fold increase in energy density during a 1-h discharge period compared to the baseline PVA/ Fe(CN)$_6^{3-/4-}$ system (Fig. 3b). Extending the discharge duration to five hours (Supplementary Figs. 51–53) further confirms the superior stability of the dual-effect TEC system, which delivers an energy density approximately twelve times higher than that of the baseline. The detailed workflow is illustrated in Fig. 3c, the device operates through three sequential stages: thermovoltage establishment under a temperature gradient, controlled discharge through an external load, and short-circuit resetting to equilibrate internal charges.

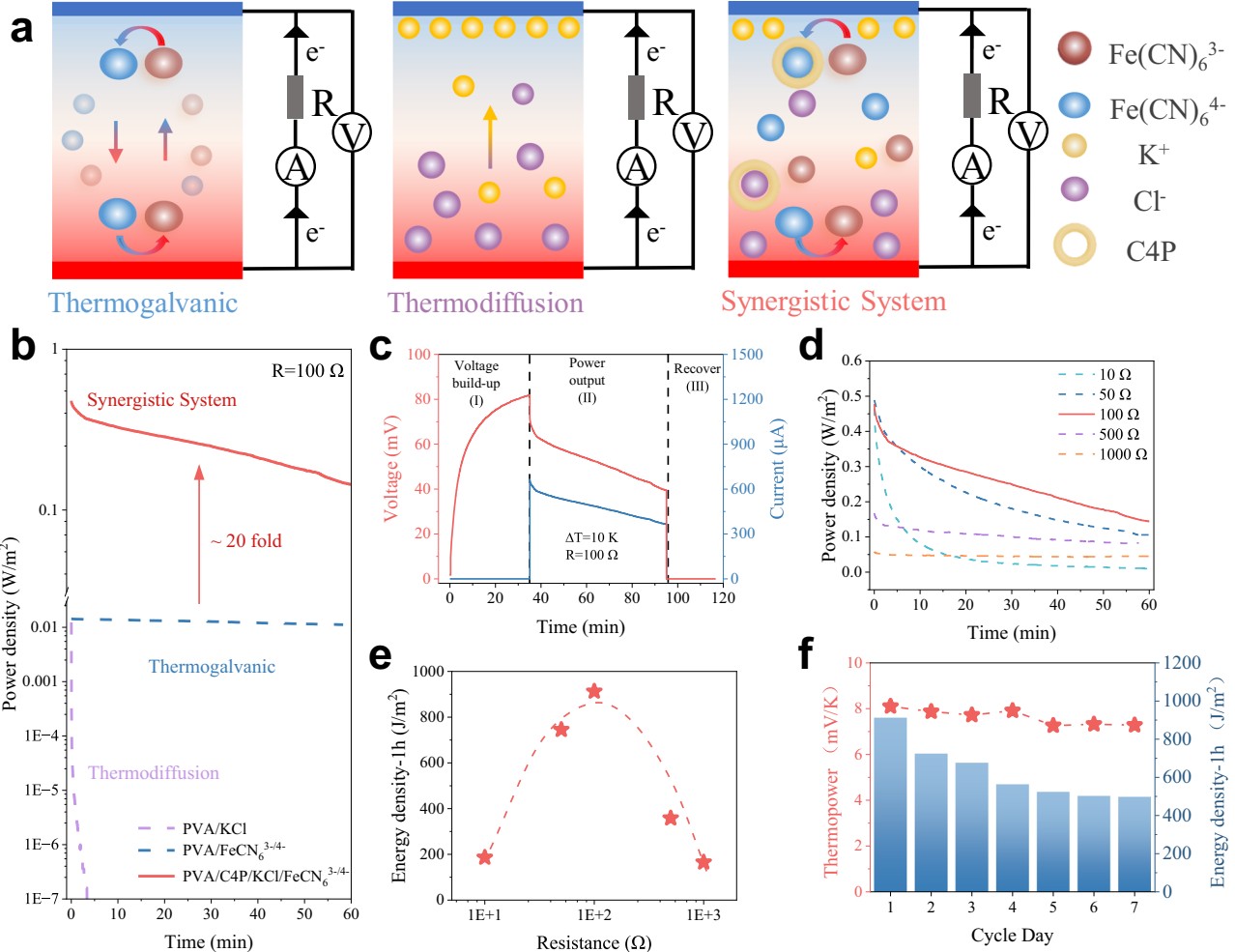

**Fig. 3 | Working modes of gel-based TECs. a** Schematic illustration of the discharge behavior for three types of thermal cells. **b** Time-dependent output power density profiles of the three cells under a temperature gradient ($\Delta T = 10$ K) during discharge for 1 h. **c** Measured voltage and current curves across three operating stages of the PVA/C4P/KCl/Fe(CN)$_6$$^{3-/4-}$ TEC. **d** Corresponding output power density of the TEC measured over 1 h during stage (II) under varioust external resistors. **e** Corresponding energy density of the TEC under different external resistances, calculated by integrating power over a 1-h discharge period. **f** Long-term performance of the TEC: energy density and thermopower retention over 7 daily discharge cycles.

Adjusting the external circuit load demonstrated that the TEC reaches a maximum energy density of 912.6 J m$^{-2}$ at an optimized load of 100 Ω (Fig. 3d, e). Over a continuous operational period of seven days, the TEC maintained approximately 90% of its initial thermopower, while its energy density retention was 54.5% (Fig. 3f and Supplementary Fig. 30). The observed performance decline after prolonged use can be primarily attributed to partial dehydration processes within the gel structure. Therefore, future studies should prioritize developing advanced encapsulation techniques and optimized storage conditions to further enhance the long-term stability and reliability of gel-based thermoelectric devices.

### Application performance of gel TECs

The mechanical performance of TECs is crucial for their practical application[30–36]. To evaluate the impact of C4P on mechanical properties, we varied C4P concentrations while keeping KCl and Fe(CN)$_6$$^{3-/4-}$ levels constant. As anticipated, the incorporation of C4P significantly enhanced the tensile properties of the gel TEC. This enhancement stems from strong hydrogen bonding between the abundant –BOH groups in C4P and –OH groups along the PVA backbone, which promotes increased cross-linking within the gel matrix (Supplementary Fig. 35). As a result, the tensile strength of the gel increased nearly threefold, reaching 406 kPa, while elongation

doubled to 410%. The improved mechanical strength enables the gel to lift weights up to 5 kg and retain over 80% of its stress capacity after 100 cycles at 100% tensile strain (Supplementary Fig. 36).

To showcase the practical application of TECs in real-world scenarios, three distinct integrated thermoelectric devices were developed, particularly focusing on wearable technology where touch and breathing represent common thermal interactions (Fig. 4a). For respiration monitoring, three elongated strips of TEC gels, each measuring $1 \times 3 \times 0.5$ cm, were affixed onto the surface of a mouthpiece, and connected in series (Fig. 4b). This configuration allows for the detection of breathing; as the wearer exhales, a temperature differential is established across the top and bottom surfaces of the gel, resulting in an observable increase in voltage. Conversely, during inhalation, the gel's surface temperature drops swiftly, leading to a rapid decrease in voltage. Additionally, we integrated three block-shaped thermoelectric gels, each measuring $1 \times 1 \times 1$ cm, in series to create an interactive human-computer interface. This assembly was designed to interface with a smartphone via a Bluetooth-enabled sensing module (Fig. 4c and Supplementary Movie 3). By applying pressure to different numbers of thermo-gel blocks, voltage signals of varying intensities are generated. This capability allows for nuanced human-computer interactions, where the intensity and pattern of touch can be translated into specific commands or inputs, showcasing

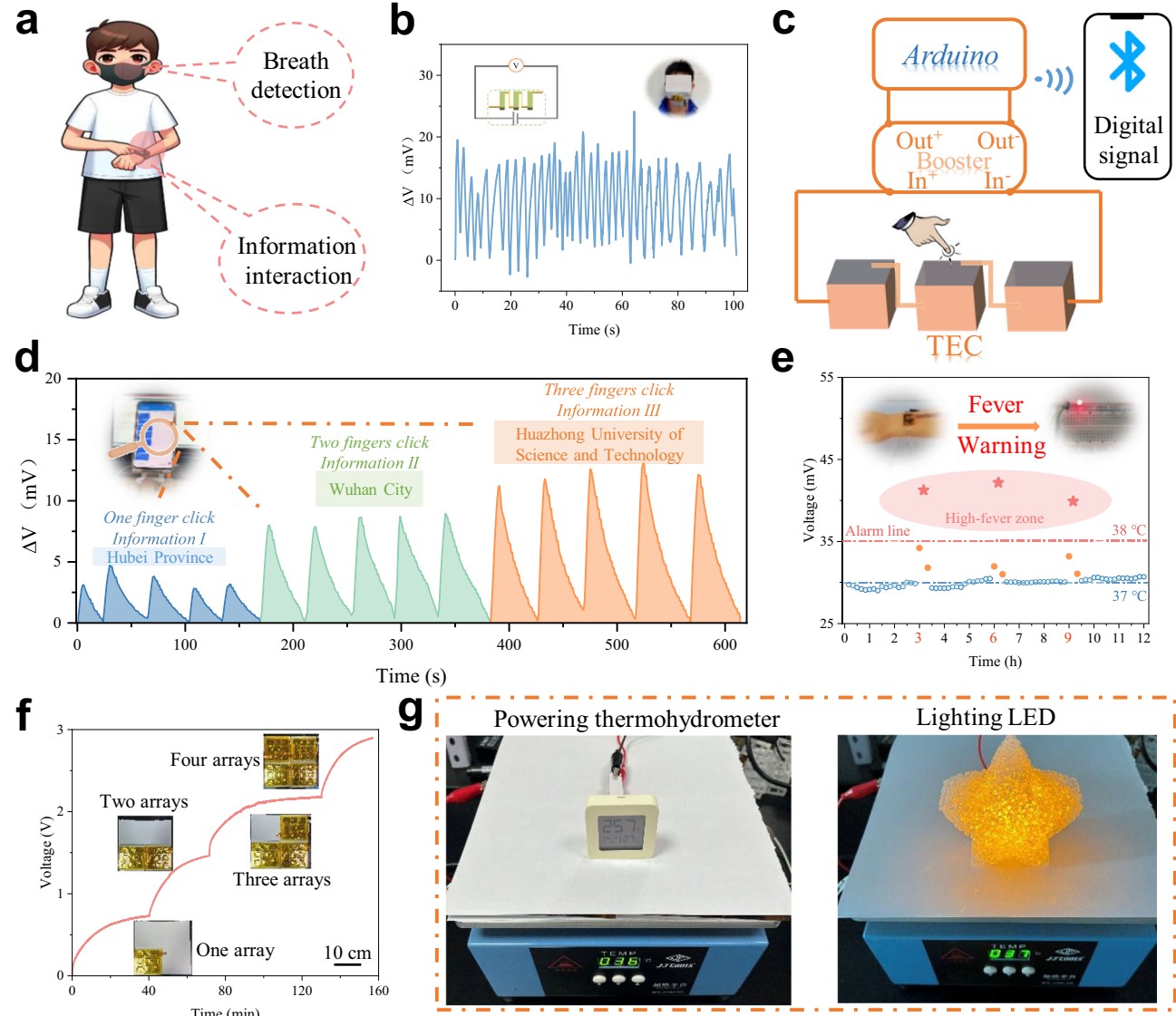

**Fig. 4 | Application demonstration of the gel TECs. a** Schematic of gel TECs in human wearable applications. **b** Voltage signals collected during the detection of human respiration by gel TECs integrated into a mask (Inset shows the device circuit diagram and optical images during testing). **c** Schematic of the circuitry of the gel TECs integrated device for human-computer interaction. **d** Voltage signals collected by the integrated interaction device during tapping (Inset is an optical image of the human-computer interaction process). **e** Body-temperature monitoring using the thermoelectric gel ($T_{hot}$ = 37–39 °C). A 2 mm gel generates $\Delta T \approx 4$ K; voltages >35 mV activate the LED. Fever events (10 min) are marked in orange. **f** Voltage-time curve ($\Delta T = 10$ K) obtained by connecting four thermoelectric converter modules in series sequentially. **g** Optical image of the four thermoelectric modules powering the thermohydrometer and red LED at a 10 K temperature difference.

the versatility and potential of TECs in creating responsive, self-powered wearable technologies (Fig. 4d). Moreover, the thermoelectric gel is well-suited for on-body applications, enabling real-time body-temperature monitoring (Fig. 4e). When attached to an artificial arm, the gel produces a stable baseline voltage under normal body-temperature (~ 37 °C). Upon simulating a fever condition (~ 39 °C), the increased thermal gradient leads to a clear voltage rise, sufficient to trigger an LED alarm. The reversible and repeatable response confirms reliable signal transduction under cyclic thermal stimuli, highlighting the potential of this soft thermoelectric system for continuous, on-body health monitoring applications.

To demonstrate the scalability and applicability of TECs, a 10 × 10 cm mold within nine gel units were arranged and internally connected in series to amplify power output, aiming to energize the device efficiently (Supplementary Fig. 37). By strategically configuring four such power generation modules in a stepped series connection, a

peak voltage of 2.9 V was realized, facilitated by a temperature differential of approximately 10 Kelvin (Fig. 4f and Supplementary Fig. 38). This configuration allows the thermoelectric cell array to directly power both a temperature/humidity meter and LEDs without any external power source (Fig. 4g). Furthermore, by harvesting waste heat rather than dissipating it, integration of the TEC with a radiative cooling film contributes to enhanced building cooling performance (Supplementary Fig. 54). Together, these results demonstrate the scalability and versatile application potential of the TEC platform.

## Discussion
In conclusion, our study introduces a approach with the integration of synergistic dual anion regulation through host-guest complexation to significantly enhance the performance of thermoelectric systems by utilizing calix[4]pyrrole within a quasi-solid-state thermocell. This approach generates a notable concentration gradient ($\Delta C$) of

$Fe(CN)_6^{4-}$ and $Fe(CN)_6^{3-}$ redox ions alongside the modulation of anions transport, culminating in a substantial enhancement of thermopower to $8.1\,mV\,K^{-1}$ and delivering a high output power density of $5.33\,mW\,m^{-2}\,K^{-2}$. This dual anion regulation mechanism not only optimizes the thermogalvanic and thermodiffusion effects but also mitigates issues related to electrolyte leakage. In addition, this study validates the technology's scalability and effectiveness under real-world conditions, underscoring its potential as a versatile solution for harnessing environmental thermal energy for diverse energy conversion applications.

## Methods

### General information

One-dimensional nuclear magnetic resonance (NMR) spectra were recorded on a Bruker AVANCE-400 spectrometer. Chemical shifts are quoted in ppm relative to TMS, which was used as an internal reference. UV–vis spectra were obtained using a UV-2600 spectrophotometer. CV measurements were performed on an electrochemical workstation (CHI760) scanning from $-1\,V$ to $+1\,V$ at $100\,mV/s$. Scanning electron microscope (SEM) images and energy dispersive X-ray spectra (EDX) were acquired on a SEM (TESCAN MIRA LMS) equipped with an EDX spectrometer. Microscopic images of the electrode sheets were acquired by a polarizing microscope (Olympus GX71). XPS was performed using a Kratos spectrometer (Thermo Scientific ESCALAB 250Xi) equipped with a monochromatic Al Kα source. XRD (Bruker D8 Advance) was used to assess the crystal structure of the synthesized samples using Cu Kα 5 ($\lambda = 1.5406\,\text{Å}$) radiation. The thermal conductivity (κ) of the samples was measured by a thermal conductivity meter (Hot Disk TPS 2500S). The electrical conductivity of the samples was obtained from the slope of the current-voltage curve from a Keithley 2450 source meter. The thermodynamic data for the complexation process were measured using an isothermal titration micro calorimeter (Malvern PEAQ ITC).

### Synthesis of the C4P

C4P was prepared according to the previous reported method (Supplementary Fig. 1)[37]. Specifically, $8.2\,g$ ($0.05\,mol$) of 4-acetylphenylboronic acid, $3.5\,mL$ ($0.05\,mol$) of pyrrole, and $3.3\,mL$ ($0.05\,mol$) of methanesulfonic acid were dissolved in methanol ($500\,mL$) and reacted under internal nitrogen atmosphere at room temperature for $5\,h$. After that, the reaction was neutralized with $NH_4OH$ and passed through a short silica gel precolumn. The volatiles were removed using a rotary evaporator and the crude product was recrystallized from acetic acid ($30-50\,mL$) to give the desired product ($5.7\,g$, 53.6% yield). $^1H$ NMR (400 MHz, dimethyl sulfoxide-d6): δ(ppm) 9.64 (4H), 7.90 (8H), 7.66(8H), 6.94 (8H), 5.88 (8H), 1.8 (12H) (Supplementary Fig. 2).

### Preparation of PVA-based thermoelectric gel cells (TECs)

The preparation process of TEC is shown in Supplementary Fig. 3. Specifically, a certain amount of KCl and C4P (this step is omitted for $PVA/Fe(CN)_6^{3-/4-}$) were added to $10\,mL$ of deionized water, followed by dissolution of KCl and dispersion of C4P using $100\,W$ ultrasonication. Subsequently, $1\,g$ ($10\,wt\%$) of PVA powder was added to the solution, and the mixture was stirred at $90\,°C$ until PVA was completely dissolved (magnetic stirrer at $500\,rpm$). The solution was then poured into silica gel molds while still warm and frozen at $-20\,°C$ for $12\,h$. After removal from the molds, the gels were left at room temperature ($\sim 25\,°C$) for $6\,h$. This freezing-thawing cycle was repeated twice more (three cycles in total) to form a PVA gel[38–41]. Finally, the obtained gels were placed in aqueous solutions of different concentrations of $Fe(CN)_6^{3-/4-}$ ($25\,°C$, $12\,h$) to obtain TECs through solvent exchange (Supplementary Figs. 10 and 11).

### Fabrication of TECs

To eliminate concerns related to electrode corrosion, hydrophilic CP (model CPM16) supplied by Shanghai TANJI Corporation was employed as the electrode material. The hydrophilic surface treatment significantly enhanced wettability without notably affecting electrical conductivity, as confirmed by contact angle and conductivity measurements (Supplementary Figs. 27, 39, and 40). The thermoelectric gel (TEC) was sandwiched between two sheets of CP and then encapsulated using polyimide (PI) tape, forming a CP|TEC|CP structured device. This configuration was used for all subsequent thermoelectric performance measurements.

### TECs performance characterization

The thermal power of the prepared TECs was measured using a homemade instrument. A flat heating table served as the heat source at the hot end, while an alumin (Al) plate with internal water circulation channels was mounted at the cold end to maintain a constant temperature through water circulation[42,43]. Unless otherwise specified, the TECs used for the tests had dimensions of $1 \times 1 \times 1\,cm$, with a spacing of $1\,cm$ between the hot and cold ends. The CP electrodes were $1.5 \times 1.5\,cm$ strips, making contact with the gel in a square area of approximately $1 \times 1\,cm$. Temperature was measured in real time by a type K thermocouple affixed to the Al plate, thereby allowing precise control of the temperature difference between the two ends of the TECs during the test. For cooling, the temperature at the hot end was rapidly reduced by activating the water circulation on the Al plate at the hot end. Voltage-time and current-voltage curves were obtained using Keithley 2450 and UNIT-61E instruments. The power-voltage curves were then calculated from the corresponding current and voltage values. A near-saturation state was defined when the measured voltage variation rate was less than $0.1\,mV\,min^{-1}$.

### Volunteer informed consent statement

The authors affirm that human research participants provided informed consent for publication of the images in Fig. 4b.

## Data availability

All data supporting the findings of this study are available in the manuscript and its Supplementary Information. All data are available from the corresponding author upon request. Source data are provided with this paper.

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

## Acknowledgements

The authors thank the Analytical & Testing Center of HUST for XRD, NMR spectroscopic, and XPS studies, and the research core facilities for life science (HUST) for ITC studies. X.C. is grateful to the National Natural Science Foundation of China (Grant No.22271110), Shenzhen Science and Technology Program (Grant No. GJHZ20240218114701003), and Natural Science Foundation of Hubei Province, China (Grant No. 2022CFA031) for financial support.

## Author contributions

X.C. conceived the project. H.L. performed experiment and data analyses. Y.L. and Y.Z. performed the synthesis, Z.G., Z.J., and J.T. performed the sensing experiments. Y.C created schematic illustrations. X.C. supervised the study. H.L. and X.C. wrote the manuscript. All authors discussed the results and commented on the manuscript.

## Competing interests

The authors declare no competing interests.
