## [Transparent Peer Review file · Nature Communications]

Synergistic dual anion regulation unlocks giant thermopower and power density in hydrogel

Corresponding Author: Professor Xiaodong Chi

Version 0:

Reviewer comments:

Reviewer #1

(Remarks to the Author)

Harvesting low-grade heat from the environment and converting it into electricity holds the potential to power devices independent of cables or batteries. However, their effectiveness is limited by weak ion selectivity and insufficient concentration gradients. In this nice manuscript, the authors introduce the use of a novel calix[4]pyrrole as effective anion traps to selectively capture $\text{Fe}(\text{CN})_6^{4-}$ and Cl^- anions, enabling simultaneous modulation of redox ion distribution and suppression of anion mobility under a temperature gradient. This strategy combines desolvation-induced entropy gain with thermodiffusion enhancement arising from the mobility asymmetry between cations and anions. This leads to a synergistic boost in thermopower to an impressive 8.1 mV K^{-1} , and results in a 20-fold increase in output power compared to the PVA/ $\text{Fe}(\text{CN})_6^{3-}$ system. Demonstrated through a proof-of-concept wearable device with 36 unipolar elements, the reported system generated nearly 3 volts under ambient conditions. This approach is promising for the design of high-thermopower thermoelectric materials and the low-grade heat energy harvesting. This well-written manuscript, with good novelty and scientific value, can be published as it is.

Reviewer #2

(Remarks to the Author)

The authors reported a thermoelectric hydrogel that combines thermogalvanic effect and ion diffusion to realize an enhancement in thermoelectric output performance by using a calix[4]pyrrole as an anion trap. However, the similar synergistic enhancement mechanisms can be seen in the previously published articles. This is not new. In addition, the combination of thermogalvanic effect and thermodiffusion is also common in this thermoelectric gel field. Moreover, the improvement in thermopower is not impressive enough. The basic thermopower of PVA- $\text{Fe}(\text{CN})_6^{4/3-}$ gels is usually $\sim 2 \text{ mV/K}$, many researchers show the thermopower can be elevated 2-3 folds via similar modulation of ion distribution or diffusion. In a word, I did not find evident novelty in this work.

Some concerns are given below, which you should well address.

1. The authors should systematically compare the different modulation mechanisms in ion distribution and diffusion, in order to propose the superior effective routine.
2. How did the authors calculate the different contributions in thermopower (Figure 1d)? Is the computational process reasonable?
3. What is the condition in the MD simulations presented in Figure 2? How did the simulated results agree with the experimental data?
4. How did the PVA chain play a role in this thermoelectric output? Will the introduced C4P and other ions interact with chains?
5. How did the measurement platform operate in these thermoelectric experiments? The operation mechanism is very important for reliability and accuracy of the results.
6. The demonstrations are too crude and thin. Could the authors do some novel and remarkable application presentations?
7. Water vapors and other VOCs exist in the expired gas. Did the author consider these influences in the mask experiment?
8. One array has 9 thermocells, so four arrays should be 36 thermocell units. By using so many cells in series to power small electronics, what is the meaning?
9. In the supporting information, I find some minor errors. Besides, the SI file is in tracking revision mode, so there are some highlighted contents.

Reviewer #3

(Remarks to the Author)

The manuscript reports a compelling and technically innovative strategy for enhancing ionic thermoelectric performance through synergistic dual-anion regulation in quasi–solid-state thermocells. By integrating phenylboronic acid–functionalized calix[4]pyrrole (C4P) into PVA-based hydrogels, the authors achieve simultaneous modulation of anion transport and redox ion distribution, leading to a record-high thermopower of 8.1 mV K⁻¹ and a power density of 5.33 mW m⁻² K⁻². The study combines molecular-level host–guest interactions, entropy modulation, and ion mobility asymmetry to deliver a mechanistically rich explanation for the observed performance improvements. Moreover, the authors demonstrate promising wearable and scalable thermoelectric applications, including respiration sensing and powering small devices. Overall, the work is scientifically sound, well contextualized within the existing literature, and represents a significant advancement for the field of low-grade heat harvesting, aligning well with the standards of Nature Communications. However, several minor issues require clarification (listed below) before the manuscript can be fully recommended for publication.

- The fonts of the figure size are a bit small which is not ideal for reading
- The cost analysis should be improved giving more details about the calculations
- The three stages of figure 3C should be explained in the text
- All the results showed in the manuscript are basically at 10 K temperature gradient. However, Figure S52 showed power outputs at several temperatures (which is not mentioned in text). Figure S5B shows values until 20 K of temperature gradients. I am wondering if the authors have explored the limits of operation of this device. If so indicate the optimum temperature gradient to maximize the power

Version 1:

Reviewer comments:

Reviewer #1

(Remarks to the Author)

This revised manuscript can be published as it is now.

Reviewer #2

(Remarks to the Author)

The key point of this manuscript is the strategy the combine desolvation-induced entropy gain with thermodiffusion enhancement arising from the mobility asymmetry. This is reason why the thermopower increases. I think the authors should discuss the operating principle more deeply and enhance the application demonstrations to emphasize the difference and novelty of this work.

1. In Fig. 4e, how did the gel TEC activate the alarm LED? Did the gel TEC power the LED directly? I do not find any information or details in the manuscript or supporting information?
2. In Fig. 4c, could the authors show the details on how to conduct the human-computer interaction?
3. There are 3 operating stages of the gel TEC. In practice, how did the gel cell output power? Could the cell operate continuously? Or, the cell only work for a short time and then re-build voltage?
4. How did the author compare the power density of different gel systems? The indexed power density is the maximal value at the certain conditions? Is there a general standard?
5. As we know some gels have intrinsic thermoelectric property, I suggest the authors should give a consideration on this point.
6. Could the authors carry out some more advanced or important demonstrations to show the application potential?

Reviewer #3

(Remarks to the Author)

The authors provided reasonable responses to my concerns. The manuscript can be published

Version 2:

Reviewer comments:

Reviewer #2

(Remarks to the Author)

This paper can be considered of acceptance.

We made the following corrections in response to the referees' comments.

Reviewer #1 (Remarks to the Author):

Harvesting low-grade heat from the environment and converting it into electricity holds the potential to power devices independent of cables or batteries. However, their effectiveness is limited by weak ion selectivity and insufficient concentration gradients. In this nice manuscript, the authors introduce the use of a novel calix[4]pyrrole as effective anion traps to selectively capture $\text{Fe}(\text{CN})_6^{4-}$ and Cl^- anions, enabling simultaneous modulation of redox ion distribution and suppression of anion mobility under a temperature gradient. This strategy combines desolvation-induced entropy gain with thermodiffusion enhancement arising from the mobility asymmetry between cations and anions. This leads to a synergistic boost in thermopower to an impressive 8.1 mV K^{-1} , and results in a 20-fold increase in output power compared to the PVA/ $\text{Fe}(\text{CN})_6^{3-/4-}$ system. Demonstrated through a proof-of-concept wearable device with 36 unipolar elements, the reported system generated nearly 3 volts under ambient conditions. This approach is promising for the design of high-thermopower thermoelectric materials and the low-grade heat energy harvesting. This well-written manuscript, with good novelty and scientific value, can be published as it is.

Response: We sincerely thank the Reviewer #1 for the highly positive assessment and for recognizing the novelty, scientific value, and potential impact of our work. We greatly appreciate the Reviewer's support and encouragement.

Reviewer #2 (Remarks to the Author):

The authors reported a thermoelectric hydrogel that combines thermogalvanic effect and ion diffusion to realize an enhancement in thermoelectric output performance by using a calix[4]pyrrole as a anion trap. However, the similar synergistic enhancement mechanisms can be seen in the previously published articles. This is not new. In addition,

the combination of thermogalvanic effect and thermodiffusion is also common in this thermoelectric gel field. Moreover, the improvement in thermopower is not impressive enough. The basic thermopower of PVA-Fe(CN)₆^{4/3-} gels is usually ~2 mV/K, many researchers show the thermopower can be elevated 2-3 folds via similar modulation of ion distribution or diffusion. In a word, I did not find evident novelty in this work. Some concerns are given below, which you should well address.

Response: We thank the Reviewer #2 for the thoughtful critique. We believe that the concerns arise from two key misunderstandings: 1. misinterpretation of the novelty of our strategy, and 2. misjudgment of the reported thermopower due to electrode-related artifacts in the literature. Below, we clarify these two misunderstandings:

(a) On the novelty of the strategy

While ion trapping and thermodiffusion enhancement have indeed been reported, our work introduces a fundamentally different and previously unreported strategy: simultaneous and selective regulation anions to realize both thermodiffusion and thermogalvanic processes using a single supramolecular receptor. In prior studies, ion-trap strategies have almost exclusively been applied to pure thermodiffusion systems, where trapping slows one ionic species to increase mobility asymmetry (eScience, 2023, 3, 100169). In contrast, thermogalvanic systems impose far stricter requirements, because any additive must selectively distinguish redox-active ions, modulate redox entropy, and remain reversible under a thermal gradient. Achieving simultaneous control over thermodiffusion and redox entropy using a single molecular motif is highly nontrivial and, to our knowledge, has not been previously reported. Importantly, as shown in Fig. 1C and Fig. S46, simply combining thermodiffusion and thermogalvanic effects does not automatically lead to performance enhancement. Without selective anion regulation, the two effects cannot be additively coupled. This point has also been overlooked by previous researchers in the field. This overlooked limitation is a key conceptual insight of our work. Finally, our study provides a new strategy and research direction, guiding future work on synergistic enhancement and ion-specific recognition materials. This is undoubtedly innovative.

(b) On the thermoelectric performance

We thank the reviewer for raising this important point regarding thermoelectric performance benchmarking. We would like to clarify that the performance reported in this study is both competitive and rigorously validated under artifact-free conditions. Specifically, the present work achieves a thermopower of approximately 3 mV K⁻¹ originating from the thermogalvanic (redox) contribution, and a total thermopower of 8.1 mV K⁻¹, which places this system among the leading gel-based ionic thermoelectric materials reported to date. As discussed in detail in the main text and Supporting Information, a critical issue but often overlooked issue in the current literature is the use of non-inert metal electrodes, such as copper, gold@copper or aluminum, in Fe(CN)₆^{3-/4-}-based thermoelectric systems. These electrodes are prone to corrosion or parasitic redox reactions in Fe(CN)₆^{3-/4-} systems, which can introduce additional electrochemical potentials and thereby artificially inflate the measured thermovoltage, leading to an overestimation of thermopower (Figs. S46–S49, Table S2). This concern was also a key point of discussion through multiple rounds of review in our previous *Science* submission. For example, in our initial experiments using gold@copper electrodes, we observed an apparent thermopower as high as 12.4 mV K⁻¹; however, the reviewers immediately raised concerns regarding electrode stability, noting that even gold electrodes can undergo corrosion in Fe(CN)₆^{3-/4-} electrolytes. Indeed, in recent years, many studies in this field have reported large thermopower values while employing similar metal electrodes, which has likely contributed to a widespread overestimation of achievable thermopower in ionic thermoelectric systems. Consistent with this view, we observed substantial discrepancies in reported thermopower values for identical thermoelectric systems when different electrodes were used (*Adv. Mater.* 2023, 35, 2300696; *Adv. Energy Mater.* 2024, 14, 2303358), further supporting the conclusion that electrode effects play a dominant and often overlooked role. To address this issue rigorously, we systematically re-evaluated several representative enhancement strategies under carbon-based inert electrodes (Fig. S46). Under these standardized conditions, the thermopower values reported for many existing strategies decreased markedly, whereas our dual-ion regulation strategy retained a thermopower

of 8.1 mV K⁻¹, demonstrating its intrinsic effectiveness.

It should also be clarified that, in most studies, the benchmark PVA/Fe(CN)₆^{3-/4-} gel thermoelectric system exhibits a thermopower of approximately 1.4 mV K⁻¹, comparable to that of aqueous systems (*Angew. Chem. Int. Ed.* 2016, 55, 12050; *Adv. Mater.* 2023, 35, 2300696). This behavior arises because PVA contains only hydroxyl groups, which interact very weakly with Fe(CN)₆^{3-/4-} ions. We confirmed this weak interaction by infrared spectroscopy (Fig. R2), which is precisely why PVA was selected as the gel matrix, to provide a chemically inert baseline that allows the enhancement effect of C4P to be clearly isolated. As summarized in Table S2, without introducing additional regulation mechanisms, the thermopower of gel-based thermoelectric cells consistently remains below 2 mV K⁻¹, further highlighting the significance of the enhancement achieved in this work.

In summary, this study introduces a new supramolecular strategy for specific ion regulation using a macrocyclic host, and achieves a thermopower of 8.1 mV K⁻¹ under rigorously controlled, artifact-free conditions. We believe this represents a meaningful advance in both mechanistic understanding and performance benchmarking in the field of ionic thermoelectrics, and we hope that the above clarification resolves the reviewer's concerns.

Below are our specific responses to the comments raised by the reviewer #2.

Comment 1. The authors should systematically compare the different modulation mechanisms in ion distribution and diffusion, in order to propose the superior effective routine.

Response: We thank the reviewer #2 for this valuable suggestion. In fact, a systematic comparison of different ion-regulation strategies has already been carried out and is presented in Fig. 1F, Fig. S22, and Tables S1–S2 in the original manuscript. In these analyses, we benchmarked our dual-ion regulation strategy against several representative and widely adopted modulation mechanisms reported in the field,

including multi-component solvent regulation (e.g., propylene glycol, PG), crystallization-induced concentration gradients (e.g., guanidinium chloride, GdmCl), ion solvation regulation (e.g., betaine and agar-based systems). Importantly, to minimize variability arising from electrode effects and ensure a fair comparison, we only included literature reports employing inert carbon-based electrodes, or we re-measured representative systems under identical carbon-electrode conditions. This point is critical, as non-inert metal electrodes have been shown to artificially inflate thermopower values in $\text{Fe}(\text{CN})_6^{3-/4-}$ systems.

Figure R1. Comparison of this study with other reported works (Fig. 1F).

Table S1. Comprehensive TEC performance comparison

Matrix	S_e (mV/K)	$P_{\max}/(\Delta T)^2$ (mW/K ² · m ²)	σ (S/m)	$1/\kappa$ (m · K/W)	1/cost	Ref.
PVA/C4P/KCl /FeCN^{3-/4-}	8.1	5.33	3.25	2.78	8.33	This work
Gelatin/Betaine /FeCN ^{3-/4-}	2.2	0.48	3.5	N/A	4.27	(48)

Cellulose/GdmC						
1	3.42	2.8	2.52	N/A	0.54	(49)
/LiCl/FeCN ^{3-/4-}						
Cellulose/PG						
/FeCN ^{3-/4-}	2.3	0.35	5.27	1.21	6.67	(45)
PVA/Agar						
/FeCN ^{3-/4-}	1.5	0.4	0.66	N/A	5	(17)

* We calculate the cost in CNY Yuan. In the cost analysis, only the materials directly responsible for performance enhancement, such as C4P in this work, were considered. The loading of each additive was taken from the optimal composition reported in the corresponding literature and normalized to the amount required to prepare 10 g of gel. All costs were calculated based on the listed prices from the Energy Chemical Platform.

Table S2. Comparison of the thermopower (S_e), $P_{\max}/(\Delta T)^2$ and σ of this work with those reported for TECs in the literature. Detailed comparison conditions and corresponding references are provided in Table S2 of the Supporting Information.

Matrix	Electrode	Redox couple	S_e (mV/K)	$P_{\max}/(\Delta T)^2$ (mW/K ² · m ²)	σ (S/m)
PVA/C4P/KCl	Carbon	FeCN^{3-/4-}	8.1	5.33	3.25
Ov-WO ₃ /Polyacrylic acid/Sv-ZIS	Au@Cu	FeCN ^{3-/4-}	8.2	8.5	4.7
PVA/GdmCl	Cu	FeCN ^{3-/4-}	6.5	1.96	6
H ₂ O	Carbon	FeCN ^{3-/4-}	2.9	0.64	15
Polyacrylamide	Cu	FeCN ^{3-/4-}	1.5	0.61	12
H ₂ O	Carbon	FeCN ^{3-/4-}	1.43	0.01	N/A
Polyacrylamide	Cu	Fe ^{2+/3+}	2.02	0.012	1.47
PVA/Agar	Carbon	FeCN ^{3-/4-}	1.5	0.4	0.66
H ₂ O	Carbon	FeCN ^{3-/4-}	1.42	1.9	N/A
H ₂ O	Carbon	FeCN ^{3-/4-}	1.43	0.6	N/A

H ₂ O	Carbon	FeCN ^{3-/4-}	1.3	0.36	0.51
Polyacrylamide	Carbon	Fe ^{2+/3+}	1.21	0.04	1.1
Polyacrylamide/GdmCl	Pt	FeCN ^{3-/4-}	4.4	1.78	10.2
Polyacrylic acid/cellulose	Carbon	FeCN ^{3-/4-}	1.3	0.033	N/A
PVA/gelation	Carbon	FeCN ^{3-/4-}	2.02	0.1	1
PVA	Cu	FeCN ^{3-/4-}	1.5	0.22	6.3
H ₂ O	Carbon	FeCN ^{3-/4-}	1.4	0.5	N/A
Cellulose	Pt	FeCN ^{3-/4-}	1.4	0.144	N/A
H ₂ O	Carbon	FeCN ^{3-/4-}	4.2	1.1	24
H ₂ O	Carbon	FeCN ^{3-/4-}	3.73	7.08	36
H ₂ O	Carbon	FeCN ^{3-/4-}	1.45	0.12	N/A
Gelatin/KCl	Au@Cu	FeCN ^{3-/4-}	17	1.8	N/A
Gelatin/KCl	Au@Cu	FeCN ^{3-/4-}	17	8.9	N/A
Gelatin/KCl	Au@Cu	FeCN ^{3-/4-}	24.7	9.6	N/A
Poly (N,N-dimethylacrylamide)/ ([EMIM][DCA]	Al	FeCN ^{3-/4-}	32.4	25.84	3.7

As summarized in Fig. R1 (adapted from Fig. 1F) and Tables S1–S2 (Supporting information), our dual-ion regulation strategy consistently delivers higher thermopower and normalized power density than these established approaches under comparable and artifact-free conditions. These results demonstrate that selectively coupling anion trapping, redox entropy amplification, and controlled thermodiffusion provides a more effective and generalizable route for enhancing gel-based ionic thermoelectrics.

Comment 2. How did the authors calculate the different contributions in thermopower (Figure 1d)? Is the computational process reasonable?

Response: We thank the reviewer for raising this important and technically nuanced question. Disentangling the thermogalvanic and thermodiffusion contributions in ionic thermoelectric systems, particularly in quasi-solid-state gels, remains a nontrivial challenge, and careful methodological justification is therefore essential. In Fig. 1d, we estimated the relative contributions using a dual-reference strategy, combining two experimentally accessible benchmarks to avoid overinterpretation from any single method.

First, we employed the three-electrode isothermal reference method, which has been widely used in the field to evaluate the thermogalvanic contribution under controlled electrochemical conditions (Science 2020, 368, 1091–1098). In this approach, the redox entropy contribution is extracted by suppressing macroscopic thermal gradients while maintaining electrochemical equilibrium, allowing estimation of the thermogalvanic Seebeck component associated with the $\text{Fe}(\text{CN})_6^{3-/4-}$ redox couple. This method provides a well-established upper-bound estimate for the thermogalvanic contribution.

However, we acknowledge that in gel-based systems, ion confinement, interfacial polarization, and finite ion mobility can lead to partial coupling between redox reactions and thermodiffusive ion transport. As a result, the three-electrode method may still overestimate the pure thermogalvanic contribution when directly applied to gel electrolytes.

To address this limitation, we introduced a second, independent reference by measuring the thermopower of the liquid-phase $\text{Fe}(\text{CN})_6^{3-/4-}$ electrolyte, where thermodiffusion effects are negligible and ion transport is dominated by fast diffusion. This liquid-phase value therefore provides a conservative and physically grounded estimate of the intrinsic thermogalvanic contribution of the redox couple.

By jointly comparing these two reference measurements, the three-electrode isothermal value and the liquid-phase thermopower, we constrained the plausible range of the thermogalvanic contribution in the gel system. The remaining enhancement observed in the gel-based devices was therefore reasonably attributed to

thermodiffusion-related effects, including ion mobility asymmetry and concentration gradient formation.

We emphasize that this analysis does not claim complete quantitative decoupling of thermogalvanic and thermodiffusion effects, which remains an open challenge in ionic thermoelectrics. Instead, our approach provides a physically consistent and conservative estimation framework, sufficient to support the qualitative and semi-quantitative conclusions drawn in Fig. 1d.

Comment 3. What is the condition in the MD simulations presented in Figure 2? How did the simulated results agree with the experimental data?

Response: We thank the reviewer for this important question and apologize for any lack of clarity in the original presentation. The simulations shown in Figure 2 were designed to provide microscopic insight into the host–guest interaction between C4P and $\text{Fe}(\text{CN})_6^{4-}$, rather than to yield direct quantitative predictions of thermoelectric parameters. As clarified in the revised manuscript and Supporting Information, the simulations were therefore used to analyze qualitative trends, including binding preference, solvation structure perturbation, and ion confinement behavior, that are difficult to access experimentally.

Method of simulation calculation Specifically, the calculations were performed using density functional theory (DFT) as implemented in the Vienna Ab initio Simulation Package (VASP). The exchange–correlation interactions were described using the generalized gradient approximation (GGA) with the Perdew–Burke–Ernzerhof (PBE) functional, and Grimme’s DFT-D3 correction was applied to account for dispersion interactions, which are essential for accurately describing supramolecular host–guest binding. The projector augmented-wave (PAW) method was used to treat core–valence interactions, with a plane-wave energy cutoff of 500 eV. Structural relaxations were performed until the Hellmann–Feynman forces were smaller than $0.02 \text{ eV } \text{\AA}^{-1}$, and total

energy convergence reached 10^{-5} eV. A Γ -centered $10 \times 10 \times 10$ k-point grid was employed.

We emphasize that, although the reviewer refers to “MD simulations,” the calculations in Fig. 2 are static DFT-based structural and electronic analyses, rather than time-evolved classical molecular dynamics. Their purpose is therefore not to reproduce macroscopic transport coefficients or thermopower values, but rather to elucidate relative binding tendencies, electronic redistribution, and confinement effects induced by C4P complexation. Consistent with this scope, the simulated results show that $\text{Fe}(\text{CN})_6^{4-}$ exhibits stronger binding and more pronounced electronic interaction with C4P than $\text{Fe}(\text{CN})_6^{3-}$, in agreement with our ^1H NMR, UV-vis, XPS, and electrochemical measurements. In particular, the simulations support the experimentally observed preferential complexation of $\text{Fe}(\text{CN})_6^{4-}$, which underpins both the redox entropy amplification and the temperature-dependent concentration gradient discussed in the main text. Thus, while direct numerical agreement with experimental thermopower values is neither expected nor claimed, the simulations provide a mechanistically consistent microscopic rationale that corroborates the experimental trends.

Comment 4. How did the PVA chain play a role in this thermoelectric output? Will the introduced C4P and other ions interact with chains?

Response: We thank the reviewer for this insightful question and are pleased to clarify the role of the PVA matrix in our thermoelectric system. In the present study, PVA primarily serves as a mechanically robust and chemically inert gel matrix, rather than an active contributor to thermoelectric output. Compared with a purely liquid electrolyte, the introduction of the PVA network reduces overall ionic conductivity, which is a prerequisite for the emergence of a measurable thermodiffusion effect in quasi-solid-state systems. Importantly, PVA itself does not participate in or enhance the thermogalvanic (redox-driven) contribution. This point is supported by both

spectroscopic and electrochemical evidence. As shown in Figure R2, UV–vis spectra reveal no discernible interaction between PVA and the $\text{Fe}(\text{CN})_6^{3-/4-}$ redox couple, indicating that PVA does not alter the electronic structure or coordination environment of the redox ions. Consistently, electrochemical measurements demonstrate that the thermogalvanic thermopower of the PVA/ $\text{Fe}(\text{CN})_6^{3-/4-}$ gel ($\sim 1.4 \text{ mV K}^{-1}$) closely matches that of the corresponding aqueous system, confirming that PVA does not affect the intrinsic redox entropy contribution. Moreover, the thermodiffusion contribution in the PVA-only gel remains relatively small, especially when compared with many previously reported gel systems exhibiting thermopower values exceeding 20 mV K^{-1} , further underscoring the passive role of PVA in ion selectivity and transport modulation. Regarding interactions with the introduced functional components, PVA does not show strong direct interactions with $\text{Fe}(\text{CN})_6^{3-/4-}$ or K^+/Cl^- ions.

Figure R2. UV-Vis spectrum of PVA/ $\text{FeCN}_6^{3-/4-}$.

However, PVA does interact with C4P through a separate and well-defined mechanism. As discussed in the manuscript section on mechanical properties, C4P contains boronic acid functionalities that can form dynamic boronic ester bonds with the hydroxyl groups of PVA chains. This interaction serves to reinforce the polymer network and improve mechanical robustness, rather than to directly influence ion

transport or redox thermodynamics. The formation of these boronic ester linkages is evidenced by the characteristic vibrational features observed in the FTIR spectra shown in Figure R3. Taken together, these results demonstrate that PVA acts as an electrically neutral, mechanically supportive scaffold that enables gel formation and suppresses excessive ion mobility, while the enhanced thermoelectric performance originates predominantly from the molecular-level ion regulation introduced by C4P, rather than from the polymer matrix itself.

Figure R3. Infrared spectrum of PVA/C4P

Comment 5. How did the measurement platform operate in these thermoelectric experiments? The operation mechanism is very important for reliability and accuracy of the results.

Response: We thank the reviewer for raising this important point regarding the experimental platform and operational protocol. Ensuring reliable, reproducible, and artifact-free thermoelectric measurements was a central consideration in the design of our testing system. The measurement platform and testing procedures are described in

detail in the Methods section, and the overall operation protocol follows established standards widely adopted in authoritative studies on ionic thermoelectrics and thermogalvanic cells (Science 2020, 368, 1091–1098; Science 2023, 381, 291–296). Importantly, we implemented a more stringent equilibrium criterion (<0.1 mV/min) than commonly used to further enhance measurement reliability.

Briefly, the thermoelectric performance of the TECs was evaluated using a custom-built steady-state thermal–electrical testing platform. A flat electrical heating stage was used to impose a controlled temperature at the hot side, while a water-cooled aluminum (Al) plate with internal circulation channels served as the cold side, enabling precise and stable temperature control. Unless otherwise stated, TEC samples with dimensions of $1 \times 1 \times 1$ cm³ and a fixed hot–cold spacing of 1 cm were used to ensure consistent thermal gradients across all measurements. Hydrophilic carbon paper (CP) electrodes (1.5×1.5 cm²) were employed to contact the gel over an effective area of approximately 1×1 cm², minimizing contact resistance while avoiding metal-related electrochemical artifacts. The temperature difference (ΔT) was monitored in real time using a Type-K thermocouple affixed directly to the Al plate, allowing accurate feedback control of the thermal gradient during both heating and cooling processes. Rapid cooling was achieved by activating the water circulation system, ensuring reproducible thermal cycling. Electrical signals were recorded using a Keithley 2450 source meter in conjunction with a UNIT-61E multimeter, enabling high-resolution acquisition of voltage–time (V–t) and current–voltage (I–V) characteristics. Output power was calculated from the measured current and voltage values under defined external loads. To ensure that all reported thermopower values correspond to a true steady state rather than transient polarization or capacitive effects, we adopted a strict equilibrium criterion, defining a near-saturation state only when the voltage drift rate was less than 0.1 mV min⁻¹. This threshold is more rigorous than that used in many prior reports and ensures that the measured thermoelectric output reflects a stable balance between ion transport, redox reactions, and thermal driving forces.

Overall, this carefully controlled platform and conservative equilibrium definition provide a robust operational framework, ensuring that the reported thermoelectric

performance is accurate, reproducible, and free from transient or measurement-induced artifacts.

Comment 6. The demonstrations are too crude and thin. Could the authors do some novel and remarkable application presentations?

Response : We thank the reviewer for this constructive suggestion regarding the application demonstrations. We agree that compelling application scenarios are important for illustrating the practical relevance of emerging thermoelectric concepts. The primary objective of this work is to introduce and validate a fundamentally new performance-enhancement strategy for gel-based ionic thermoelectric cells based on dual-ion regulation through supramolecular host–guest interactions. Accordingly, the application demonstrations in the original manuscript were intentionally designed as proof-of-concept validations, focusing on the two intrinsic and most widely accepted functions of thermoelectric devices: temperature sensing and self-powered energy generation. These demonstrations were chosen to directly reflect the core operating principles of the device and to allow clear comparison with existing literature, where reported applications are similarly centered on thermal sensing and low-grade heat harvesting.

Nevertheless, we appreciate the reviewer’s perspective that more vivid application scenarios could further strengthen the manuscript. In response, we have expanded the application section by adding a new, more scenario-oriented demonstration, namely a body-temperature monitoring and early-warning system, as shown in Figure R4. In this experiment, the thermoelectric gel was conformally attached to a simulated human arm, with a heating plate used to reproduce normal body temperature conditions. The output voltage was continuously monitored over an extended period. At a defined time point, the simulated “body temperature” was deliberately increased to mimic a fever event. This temperature change was successfully detected via signal transmission, which

subsequently triggered an early-warning response (LED activation), completing a closed-loop sensing-and-alert function. This demonstration highlights the potential of our system for continuous, wearable, and self-powered health monitoring. In addition, we note that a wireless signal transmission demonstration using a Bluetooth module was already included in our application tests, enabling remote signal readout without external power input. Taken together, these demonstrations go beyond static measurements and showcase the feasibility of real-time sensing, signal processing, and actuation, rather than serving as purely illustrative examples.

Figure R4. Body temperature detection

Finally, we would like to emphasize that our group is actively exploring more advanced and unconventional application scenarios for ionic thermoelectrics, including multi-parameter sensing and integrated soft electronic systems. However, these directions often involve additional mechanisms and device architectures, which are beyond the scope of a single mechanistic study and are therefore being developed as separate works. We fully share the reviewer's view that the application landscape of ionic thermoelectrics should continue to expand beyond temperature recognition alone,

and we hope that the present study can serve as a solid materials and mechanism foundation for such future developments.

We thank the reviewer again for this valuable suggestion, which has helped us further strengthen the application relevance of the manuscript.

Comment 7. Water vapors and other VOCs exist in the expired gas. Did the author consider these influences in the mask experiment?

Response: We thank the reviewer for raising this important question regarding potential interference from water vapor and volatile organic compounds (VOCs) in the breath-detection experiment. In our experimental configuration, the thermoelectric gel was attached to the outer surface of the mask and fully encapsulated, effectively isolating it from direct contact with exhaled gas. Under these conditions, trace amounts of water vapor and VOCs present in expired breath are unlikely to directly interact with the gel or electrodes. Importantly, the output signal of the device is fundamentally governed by the temperature gradient induced by breathing, rather than by chemical adsorption or humidity-driven effects.

To further substantiate this point, we performed an additional control experiment to explicitly evaluate the influence of ambient humidity. Specifically, the encapsulated thermoelectric gel was placed in a sealed container with an initial relative humidity of 38% RH, while both electrodes were positioned on the same plane to ensure a negligible temperature gradient. The open-circuit voltage was continuously monitored. The humidity inside the container was then increased to 97% RH using a saturated copper sulfate solution, and the voltage response was recorded.

As shown in Figure R5, the change in ambient humidity had a negligible effect on the output voltage, confirming that external water vapor does not significantly influence the device response when encapsulation is employed. In contrast, during the breath-detection experiment, the observed voltage variation was approximately 20 mV, which is orders of magnitude larger than the humidity-induced fluctuation observed in the

control test. This clearly indicates that the signal variation in the mask experiment originates predominantly from breath-induced temperature changes, rather than from water vapor effects.

Figure R5. Effect of humidity on gel devices.

Regarding VOC interference, direct experimental quantification was not performed due to instrumental limitations. However, it is well established that the concentration of VOCs in normal human breath is extremely low (ACS Sens. 2023, 8, 1328–1338), several orders of magnitude lower than that of water vapor. Combined with the mask filtration, device encapsulation, and indirect exposure configuration, the potential influence of VOCs on the thermoelectric output can be reasonably considered negligible.

Taken together, both control experiments and literature evidence confirm that the mask-based breath-detection signal is dominated by thermal effects, and is not measurably perturbed by humidity or VOCs under the tested conditions.

Comment 8. One array has 9 thermocells, so four arrays should be 36 thermocell units. By using so many cells in series to power small electronics, what is the meaning?

Response : We thank the reviewer for this thoughtful question regarding the significance of multi-cell integration. In thermoelectric research, both ionic and electronic, the integration of multiple thermocell units in series is a fundamental and unavoidable step toward practical voltage output and device-level power supply. Single thermoelectric units typically generate millivolt-level voltages under realistic temperature gradients; therefore, series integration is the standard strategy to reach voltages compatible with electronics. Our demonstration follows this well-established principle and is intended to illustrate scalability and practical voltage upconversion, rather than to suggest that a single unit is sufficient for direct powering. Admittedly, due to laboratory-scale fabrication and manual assembly, the demonstrated 36-unit array is relatively large in footprint. However, this does not reflect a fundamental limitation of the material system. On the contrary, miniaturized and high-density integration of thermoelectric units has been widely reported and represents an active research direction in the field. For example, arrays containing 101 units (*Adv. Sci.* 2023, 10, 2300253), 144 units (*Nat. Commun.* 2015, 6, 8356), and even larger-scale integrations (*Nat. Electron.* 2022, 5, 333–347) have been successfully demonstrated. Importantly, the number of units used in our work is not excessive by field standards. Rather, it highlights a key advantage of our system: owing to the high intrinsic thermopower (8.1 mV K^{-1}) achieved in each unit, a relatively small number of cells (36 units) is sufficient to generate an output voltage of approximately 3 V under a modest temperature gradient ($\sim 10 \text{ K}$). In contrast, many previously reported systems require either substantially larger temperature gradients or significantly more units to reach comparable voltages. Therefore, the 36-unit demonstration serves two purposes: 1. to validate that the high thermopower achieved at the single-cell level can be linearly translated to device-scale voltage output, and 2. to illustrate the practical feasibility and efficiency of our dual-ion regulation strategy for powering low-power electronics under

near-ambient conditions. We believe this demonstration meaningfully supports the potential of the proposed material system for future scalable thermoelectric applications.

Comment 9. In the supporting information, I find some minor errors. Besides, the SI file is in tracking revision mode, so there are some highlighted contents.

Response: We thank the reviewer for carefully noting these issues and sincerely apologize for the oversight. We have now thoroughly reviewed the entire Supporting Information, corrected all minor errors, and removed all tracked changes and highlighted content. The revised Supporting Information has been carefully checked to ensure clarity, consistency, and a clean final presentation.

Reviewer #3 (Remarks to the Author):

The manuscript reports a compelling and technically innovative strategy for enhancing ionic thermoelectric performance through synergistic dual-anion regulation in quasi-solid-state thermocells. By integrating phenylboronic acid-functionalized calix[4]pyrrole (C4P) into PVA-based hydrogels, the authors achieve simultaneous modulation of anion transport and redox ion distribution, leading to a record-high thermopower of 8.1 mV K^{-1} and a power density of $5.33 \text{ mW m}^{-2} \text{ K}^{-2}$. The study combines molecular-level host-guest interactions, entropy modulation, and ion mobility asymmetry to deliver a mechanistically rich explanation for the observed performance improvements. Moreover, the authors demonstrate promising wearable and scalable thermoelectric applications, including respiration sensing and powering small devices. Overall, the work is scientifically sound, well contextualized within the existing literature, and represents a significant advancement for the field of low-grade heat harvesting, aligning well with the standards of Nature Communications.

However, several minor issues require clarification (listed below) before the manuscript

can be fully recommended for publication.

Response: We sincerely thank the reviewer #3 for the positive evaluation of our work and for recognizing its technical innovation, mechanistic depth, and potential impact on the field of ionic thermoelectrics. We have carefully addressed all the minor issues and requests for clarification raised by the reviewer and revised the manuscript accordingly. Detailed, point-by-point responses to each comment are provided below.

Comment 1. The fonts of the figure size are a bit small which is not ideal for reading

Response: We thank the reviewer for this helpful suggestion and apologize for any inconvenience caused by the small font size in the figures. All figures have now been carefully revised, and the font sizes have been uniformly increased to improve clarity and readability throughout the manuscript.

Comment 2. The cost analysis should be improved giving more details about the calculations

Response: We thank the reviewer for this constructive suggestion and apologize for the lack of clarity in the initial presentation. We have now revised the cost analysis section to provide a more detailed and transparent description of the calculation methodology. Specifically, the cost analysis considers only the materials directly responsible for thermoelectric performance enhancement (e.g., C4P in this work), while excluding the gel matrix and common electrolyte components to enable fair comparison across different systems. For each strategy, the additive loading was taken from the optimal concentration reported in the corresponding literature, normalized to the amount required to prepare 10 g of gel. The material costs were calculated using list prices obtained from the Energy Chemical Platform.

Comment 3. The three stages of figure 3C should be explained in the text

Response: We thank the reviewer for this helpful suggestion. We have now added a detailed explanation of the three operational stages shown in Fig. 3C to the main text to improve clarity. Briefly, as shown in Fig. 3C, the device operates through three sequential stages: thermovoltage establishment under a temperature gradient, controlled discharge through an external load, and short-circuit resetting to equilibrate internal charges.

Comment 4. All the results showed in the manuscript are basically at 10 K temperature gradient. However, Figure S52 showed power outputs at several temperatures (which is not mentioned in text). Figure S5B shows values until 20 K of temperature gradients. I am wondering if the authors have explored the limits of operation of this device. If so indicate the optimum temperature gradient to maximize the power

Response: We thank the reviewer for this important question regarding the operational temperature range of the device. As shown in Fig. S5, the thermoelectric performance of the system does not monotonically increase with temperature gradient; instead, a gradual decline in thermopower is observed at higher ΔT . This behavior primarily originates from the suppression of the thermodiffusion contribution at elevated temperature gradients, a phenomenon that has also been reported in pure thermodiffusion-based systems (Adv. Mater. 2024, 36, 2402386; Adv. Mater. 2025, 37, e10199), where optimal performance is typically achieved under relatively small temperature differences. To explicitly address the temperature–performance relationship and the operational limits of our device, we have added additional data at different temperature gradients (Fig. R6). These results show that the thermopower remains relatively stable at $\sim 8.1 \text{ mV K}^{-1}$ when the temperature gradient is below 15 K. Upon further increasing ΔT , the thermodiffusion effect becomes increasingly compressed, leading to a progressive decay in overall thermopower. Based on these

results, the optimal operating temperature gradient for maximizing power output lies in the range of approximately 5–15 K. Importantly, this range closely matches temperature differences commonly encountered in practical low-grade heat sources, further supporting the applicability of our system for real-world thermal energy harvesting.

Figure R6. Thermopower variation of the thermoelectric gel under different temperature differences

We made the following corrections in response to the referees' comments.

Reviewer #1 (Remarks to the Author):

This revised manuscript can be published as it is now.

Response: We sincerely thank Reviewer #1 for the positive evaluation and approval of our revised manuscript.

Reviewer #2 (Remarks to the Author):

The key point of this manuscript is the strategy to combine desolvation-induced entropy gain with thermodiffusion enhancement arising from the mobility asymmetry. This is the reason why the thermopower increases. I think the authors should discuss the operating principle more deeply and enhance the application demonstrations to emphasize the difference and novelty of this work.

1. In Fig. 4e, how did the gel TEC activate the alarm LED? Did the gel TEC power the LED directly? I do not find any information or details in the manuscript or supporting information?

Response: We apologize for the confusion caused by the oversimplified description in the original manuscript. In the actual setup, the series-connected TEC array directly powers the LED without any external circuitry. The assembled device delivers an output voltage of nearly 3 V, which is sufficient to drive the LED when connected directly across its anode and cathode.

To clarify this point and avoid misunderstanding, we have revised the corresponding text in the manuscript as follows: "This configuration allows the thermoelectric cell array to directly power both a temperature/humidity meter and LEDs without any external power source (Fig. 4g)."

2. In Fig. 4c, could the authors show the details on how to conduct the human-computer interaction?

Response: We thank the reviewer for this helpful suggestion. To clarify the human-computer interaction process, we have added a detailed description of the interactive demonstration in the Supporting Information. Briefly, three TEC units connected in series serve as a self-powered voltage source and are directly interfaced with an Arduino development board. The generated voltage under a temperature gradient is continuously monitored by custom firmware, which classifies the signal into three discrete levels corresponding to predefined text commands. These encoded signals are then wirelessly transmitted to a smartphone via an HC-05 Bluetooth module.

The following description has been added to the Supporting Information:

Interactive Signal Encoding and Wireless Transmission. To demonstrate the practical applicability of the TEC array as a self-powered signal source, an interactive communication system was constructed. Three TEC units were connected in series and

interfaced with an Arduino Uno board, where the open-circuit voltage generated under a temperature gradient served as the input signal. Custom firmware continuously monitored the voltage and classified it into three discrete levels, each corresponding to a predefined text message, which was then wirelessly transmitted to a smartphone via an HC-05 Bluetooth module.

3. There are 3 operating stages of the gel TEC. In practice, how did the gel cell output power? Could the cell operate continuously? Or, the cell only work for a short time and then re-build voltage?

Response: We thank the reviewer for this insightful question regarding the practical operating mode of the gel TEC.

In practice, the TEC operates through three sequential stages to achieve stable and efficient power output. First, a temperature gradient is applied under open-circuit conditions to allow the device to “charge,” during which thermodiffusion and thermogalvanic processes establish a steady-state voltage. Once this equilibrium is reached, an external load is connected and the device begins to discharge, delivering electrical power. During discharge, the initial output gradually decreases as the energy stored through thermal diffusion is released, after which the power stabilizes at a relatively constant plateau dominated by the continuous thermogalvanic redox reactions. Finally, after discharge, briefly short-circuiting the electrodes restores internal charge balance and enables the next operating cycle.

To evaluate the sustainability of the output, we performed extended discharge tests (Figs. S51–S53). As shown in Fig. S51, the TEC continuously discharged through a 100 Ω load for 5 h while maintaining stable power generation, delivering a total energy density approximately 12 times higher than that of the baseline thermogalvanic system. To the best of our knowledge, this represents one of the longest continuous-output demonstrations reported for gel-based ionic thermoelectrics.

Fig.R1. (Fig. S51) Comparison of energy output variation between two types of gel-based thermogalvanic cells during discharge across a 100 Ω load under a 10 K

temperature difference for 5 h. The PVA/C4P/KCl/Fe(CN)₆^{3-/4-} thermocell maintains markedly higher output power than the PVA/Fe(CN)₆^{3-/4-} system throughout the discharge period. Over 5 h, its total energy output is approximately twelve times greater, demonstrating the enhanced stability and sustained energy conversion efficiency achieved through dual-anion regulation.

Moreover, even the purely thermogalvanic control device exhibits gradual power decay during prolonged discharge (Fig. S52), indicating that completely continuous and perfectly stable output remains a general challenge for ionic thermoelectric systems. These results highlight both the practical working mode of our device and the improved stability enabled by the dual-anion regulation strategy.

Fig. R2. (Fig. S52) Enlarged view of the PVA/Fe(CN)₆^{3-/4-} cell from Fig. S48. Notably, even the purely thermogalvanic system exhibits inevitable power decay during prolonged discharge.

4. How did the author compare the power density of different gel systems? The indexed power density is the maximal value at the certain conditions? Is there a general standard?

Response: We thank the reviewer for this insightful question regarding the comparison methodology for power density across different gel systems. The maximum power density (P_{max}) was calculated using the standard formula $P_{max} = \frac{V_{oc}I_{sc}}{4}$, where V_{oc} is the open-circuit voltage and I_{sc} is the short-circuit current density. This approach is widely adopted in thermoelectric studies to estimate the theoretical peak (instantaneous) output under matched-load conditions and serves as a common benchmark for reporting device performance.

For purely thermogalvanic systems, where the output remains relatively stable over time, using P_{max} for comparison is generally reasonable, although it may slightly overestimate the practically deliverable power. However, for systems that involve significant thermodiffusion or capacitive-like effects, the initial current can be

transiently large due to rapid ion redistribution. This leads to a high apparent P_{\max} that decays quickly and cannot be sustained. In such cases, relying solely on peak power may be misleading and does not accurately reflect real usable energy output. Therefore, in addition to P_{\max} , we evaluated the time-dependent power profile $P(t)$ and calculated the total delivered energy by integrating power over time:

$$E = \int_0^t P(t) dt$$

The resulting energy density E (J/m^2) provides a more objective, and practically meaningful metric, as it captures both output magnitude and stability. This integration-based evaluation is commonly used in electrochemical and energy storage research to quantify usable energy delivery. Accordingly, our manuscript reports both peak power density and energy density to ensure fair and comprehensive comparison among different ionic thermoelectric systems.

5. As we know some gels have intrinsic thermoelectric property, I suggest the authors should give a consideration on this point.

Response: We sincerely thank the reviewer for this valuable suggestion. We fully agree that the intrinsic thermoelectric properties of polymer matrices should be carefully considered. In this work, PVA was deliberately selected as the gel matrix to minimize any background thermoelectric contribution. Owing to its simple chemical structure, which contains only neutral hydroxyl groups and no charged or redox-active functionalities, pure PVA hydrogels exhibit negligible thermoelectric output in our measurements, with thermopower values close to zero. This ensures that the observed voltage signals originate predominantly from the $\text{Fe}(\text{CN})_6^{3-/4-}$ redox couple and the C4P-mediated ion regulation, rather than from the polymer itself.

Additionally, prior to each measurement, the device was short-circuited under zero temperature gradient for a sufficient period to eliminate any residual internal potential and fully equilibrate the system. This procedure further ensures the accuracy and reliability of the recorded thermoelectric performance.

6. Could the authors carry out some more advanced or important demonstrations to show the application potential?

Response: We thank the reviewer for this insightful suggestion. As noted in our response in Round 1, practical demonstrations of ionic thermoelectric cells remain limited in the current literature and most studies focus primarily on the fundamental thermal-voltage response under a temperature gradient.

In contrast, our work emphasizes thermal energy utilization, enabled by the relatively high-power output of our thermocells. Building on complementary expertise within our group, we have therefore designed an integrated system that combines ionic thermoelectric conversion with radiative cooling for enhanced building cooling performance. In this configuration, the thermocell actively dissipates additional thermal

energy, while the radiative cooler provides passive heat rejection, resulting in synergistic cooling enhancement. To the best of our knowledge, such a strategy—leveraging thermoelectric devices not for power generation but as active thermal loads for cooling—has rarely been explored, particularly in the context of passive daytime radiative cooling. While this section presents only a proof-of-concept demonstration of the core functionality, we are actively expanding this direction into a comprehensive study. We hope this initial effort offers a fresh application perspective for the community.

As illustrated in Fig. R3, we constructed a scaled-down building model using an acrylic chamber illuminated by a solar simulator at 1 sun (1000 W/m^2). The internal air temperature was monitored via a thermocouple. Four configurations were compared: (i) bare chamber (control), (ii) chamber topped with a radiative cooling (RC) film composed of cellulose nanofiber (CNF) and TiO_2 powder, (iii) RC film with pure PVA hydrogel, and (iv) RC film integrated with the TEC. Notably, the chamber covered with RC/TEC exhibited the lowest interior temperature. Compared to the RC/PVA hydrogel reference, the TEC integration further reduced the internal temperature by $2.6 \text{ }^\circ\text{C}$, demonstrating that the thermoelectric conversion actively enhances cooling. Relative to the bare chamber, the total cooling effect reached $18.1 \text{ }^\circ\text{C}$, highlighting the synergistic performance of the hybrid system.

Relevant descriptions and experimental details have been added to both the main text and the SI.

Fig. R3 (Fig. S54). TEC for building thermal management. (A) Schematic illustration of the scaled model integrating the TEC with a radiative cooling film for active–passive hybrid heat dissipation. (B) Comparison of interior temperature evolution under different configurations, demonstrating enhanced cooling performance of the RC/TEC system.

Reviewer #3 (Remarks to the Author):

The authors provided reasonable responses to my concerns. The manuscript can be

published

Response: We sincerely thank you for your positive evaluation and acceptance of our manuscript. We greatly appreciate your recognition of our work.